# Long-Term Prognostic Value of Myocardial Viability by Myocardial Contrast Echocardiography in Patients after Acute Myocardial Infarction: A Systematic Review and Meta-Analysis

**DOI:** 10.3390/medicina58101429

**Published:** 2022-10-11

**Authors:** Jingxin Wang, Mengxi Yang, Zhi Yang, Lu Ye, Hong Luo, Yingkun Guo

**Affiliations:** 1Department of Ultrasound, Key Laboratory of Birth Defects and Related Diseases of Women and Children of Ministry of Education, West China Second University Hospital, Sichuan University, Chengdu 610041, China; 2Department of Radiology, State Key Laboratory of Biotherapy and Cancer Center, West China Hospital, Sichuan University, Chengdu 610041, China; 3Department of Radiology, Chengdu Fifth People’s Hospital, Chengdu 611130, China; 4Department of Radiology, Key Laboratory of Birth Defects and Related Diseases of Women and Children of Ministry of Education, West China Second University Hospital, Sichuan University, 20# Section 3 South Renmin Road, Chengdu 610041, China

**Keywords:** myocardial contrast echocardiography, acute myocardial infarction, myocardial viability, prognosis, meta-analysis

## Abstract

*Background and Objectives:* According to recent guidelines, myocardial contrast echocardiography (MCE) is recommended for detecting residual myocardial viability (MV). However, the long-term prognostic value of MV as assessed by MCE in identifying major adverse cardiac events (MACE) after acute myocardial infarction (AMI) remains undefined. *Materials and Methods:* We searched multiple databases, including PubMed, EMBASE, and Web of Science for studies on the prognostic value of MCE for clinical outcomes in AMI patients. The primary endpoints were MACEs during follow-up. Six studies that evaluated a total of 536 patients with a mean follow-up of 36.8 months were reviewed. *Results:* The pooled sensitivity and specificity of MCE for predicting MACEs were 0.80 and 0.78, respectively, and the summary operating receiver characteristics achieved an area under the curve of 0.84. The pooled relative risks demonstrated that the MV evaluated by MCE after AMI was correlated with a high risk for total cardiac events (pooled relative risk: 2.07; 95% confidence interval: 1.28–3.37) and cardiac death (pooled relative risk: 2.48; 95% confidence interval: 1.03–5.96). MV evaluated by MCE was a highly independent predictor of total cardiac events (pooled hazard ratio: 2.09, 95% confidence interval: 1.14–3.81) in patients after AMI. *Conclusions:* Residual MV evaluated by MCE may be an effective long-term prognostic tool for predicting MACE in patients after AMI that can provide moderate predictive accuracy. The assessment of MV by MCE may become an alternative technique with the potential to rapidly provide important information for improving long-term risk stratification in patients after AMI, at the bedside in clinical practice, especially for patients who cannot tolerate prolonged examinations. The PROSPERO registration number is CRD42020167565.

## 1. Introduction

Acute myocardial infarction (AMI) continues to be a significant global public health concern [1,2,3]. Although the mortality rate of patients with AMI has decreased, heart failure secondary to myocardial infarction remains a major cause of morbidity and mortality [4,5]. Therefore, risk stratification in patients after AMI is crucial to guide treatment strategies and improve long-term outcomes [5,6].

The purpose of treatment after AMI, regardless of whether the treatment is thrombolysis or emergency percutaneous coronary intervention (PCI), is to reopen the infarct-related artery [7]. However, even with a successful recanalization of criminal vessels, some patients still suffer unsuccessful myocardial reperfusion. This phenomenon is known as no-reflow or low-reflow. Coronary angiography only reflects the condition of the epicardial coronary artery. Myocardial contrast echocardiography (MCE) can reflect the condition of myocardial microcirculation by myocardial perfusion, which can be used reliably to assess infarct size and hence myocardial viability (MV) [8,9,10]. Therefore, MCE may be used as an effective bedside method to evaluate the efficacy of coronary revascularization after AMI, which is of great value in assessing MV. This is important as it could potentially guide treatment, evaluate cardioprotective therapies, and improve clinical outcomes [6].

A multicenter Acute Myocardial Infarction Contrast Imaging study demonstrated that, among patients with TIMI 3 flow after ST elevation myocardial infarction, the extent of microvascular damage assessed by MCE is the most powerful independent predictor of left ventricle remodeling compared to other established clinical and angiographic parameters of reperfusion [11]. Recently, several studies [12,13,14,15,16,17] have further explored the long-term prognostic significance of MCE in patients with AMI. However, the clinical significance of these studies has been limited by small sample sizes, single-center cohorts, and inconsistent results. To date, no meta-analysis has been conducted to clarify and address this issue. Therefore, we performed a systematic review and meta-analysis of these studies to evaluate and summarize the long-term prognostic value of MV, as assessed by MCE, after AMI.

## 2. Methods

Our main review and assessment processes were conducted in accordance with the Preferred Reporting Items for Systematic Reviews and Meta-Analyses standards [18] and are described below.

The study protocol was prospectively registered in the PROSPERO database (CRD42020167565).

### 2.1. Search Strategy

We searched for eligible studies using the PubMed database, EMBASE database, and Web of Science with various combinations of free-text words and MeSH subheadings, including “myocardial”, “contrast echocardiography”, “infarct*”, “event*”, ‘‘prognosis*”, “predict*”, and “diagnosis*”. We also performed a manual search of relevant publications to identify additional eligible trials. The search analyzed original literature published in such databases up to 1 June 2022, with no date restrictions. The detailed methods we used to search PubMed and EMBASE are shown in Appendix A.

### 2.2. Selection Criteria

The inclusion criteria in this meta-analysis were as follows: (1) prospective study, (2) population: post-ST elevation AMI patients, (3) underwent MCE, (4) semiquantitative scoring of MCE findings, (5) at least 6 months follow-up, and (6) full-text in the English language. The endpoint was MACE, which included total cardiac events and cardiac death.

Studies were excluded if they were (1) published as crossover studies, retrospective studies, comments, letters, editorials, case reports, or reviews; (2) used in vitro or animal models. If two or more studies used the same set of main experimental data, we selected the one that was the most informative.

### 2.3. Study Collection and Data Extraction

Two physician-investigators independently conducted the literature search and extracted potentially relevant studies in a standardized manner. A third reviewer reviewed the data. Disagreements were resolved by discussion and by referral of the study to a fourth reviewer.

We extracted the following demographic data from the studies: author, year of publication, study population (sample size, age, and sex of subjects), study design, presence of coronary risk factors (diabetes mellitus, hypertension, dyslipidemia, family history of coronary artery disease, and smoking status), and follow-up duration. We also retrieved the technical characteristics of MCE, criteria of the semiquantitative scoring system, inclusion/exclusion criteria, treatment, and primary endpoints. On full-text review, we extracted hazard ratios (HRs) or odds ratios (ORs) with corresponding 95% confidence intervals (CIs) for MACE occurrence among post-AMI patients. In case of disagreement among data extractors, the final decision was made by a consensus of all authors.

### 2.4. Quality Assessment

The Newcastle–Ottawa Quality Assessment Scale (0–9 points) was used by two independent reviewers to evaluate study quality [19]. The following criteria were assessed for each study: (1) selection of the study groups, (2) comparability between groups, and (3) outcomes. In addition, we evaluated important quality metrics on the prognostic factor as proposed by Hayden et al., [20] including a clear definition of the prognostic factor, appropriate cutoff points, adequate validity of prognostic factor measurement, and adequate analysis. A third reviewer was consulted in case of any disagreements.

### 2.5. Statistical Analysis

Statistical analysis was performed using Review Manager (RevMan) V5.3 (Cochrane Information Management System, Oxford, UK) and Stata (StataCorp, College Station, TX, USA). The inconsistency index (*I*^2^) was used to determine homogeneity. A random-effects model was used in the absence of unexplained statistical heterogeneity (*I*^2^ > 50%). Chi-square statistics were used to assess the magnitude of the heterogeneity. Pooled sensitivity, specificity, and diagnostic OR were computed. A summary receiver-operating characteristic (ROC) curve was constructed to calculate the area under the curve (AUC), which was used as a global measure of test performance. The combined effect and pooled adjusted relative risk (RR) (including HR and OR) with 95% CI were computed from all the studies. RR > 1, RR = 1, and RR < 1 indicated that patients with abnormal MV might have had a higher, similar, or lower risk of MACE after AMI compared to patients with normal MV, respectively. The model used to calculate pooled RRs with 95% CI was based on the *I*^2^ statistic (presence or absence of heterogeneity). Finally, a sensitivity analysis (leave-one-out approach) was conducted to explore heterogeneity among individual studies; the pooled RRs were recalculated by leaving one study out at a time. Statistical significance for hypothesis testing was set at a two-tailed *p*-value of 0.05.

## 3. Results

This section may be divided by subheadings. It should provide a concise and precise description of the experimental results, their interpretation, as well as the experimental conclusions that can be drawn.

### 3.1. Systematic Review

Our literature search initially identified 322 articles; of these, 142 irrelevant articles were excluded based on the title and abstract review. A further 46 articles were excluded on full-text review for various reasons, including the absence of semiquantitative analysis or a lack of pre-specified outcomes. At last, six studies were eventually used for a detailed study [13,14,16,17,21,22]. A flowchart of our search results is presented in Figure 1.

The baseline characteristics and the pooled characteristics of these studies are shown in Table 1. In total, there were 536 patients across the studies, ranging from 27 to 167 patients per study. Demographically, 73.8% of the subjects were male, and their ages ranged from 47 to 78 years. The follow-up period ranged from 10.8 to 78 months, with an average of 36.8 months.

Table 1 also shows the inclusion and exclusion criteria, treatment, and primary endpoints evaluated in each study. All six studies were prospective studies of post-AMI patients. Cardiac events were evaluated during regular follow-up visits. Table 2 summarizes the number of patients with MACE and the outcomes of the multivariate analysis of risk for MACEs.

### 3.2. Characteristics of MCE and Semiquantitative Systems

The characteristics of the MCE and semiquantitative systems are summarized in Table 3.

Each study named the semiquantitative prognostic parameter differently, including contrast defect index (CDI), contrast score index (CSI), regional perfusion score index (RPSI), and perfusion score index (PSI). However, all of them were calculated in the same way: the sum of MCE scores in each segment was divided by the total number of segments. Thus, we referred to the abovementioned index as the MCE score index in this meta-analysis. Similarly, the definition of a semiquantitative system was almost uniform in these six articles, defined on a three-point scale: homogenous contrast perfusion, partial/patchy contrast perfusion, and absent contrast perfusion.

### 3.3. Ability of MCE Score Index to Predict MACE in Post-AMI Patients

This meta-analysis assessed the ability of the MCE score index to predict MACE and summarized the parameters used to predict outcomes. Four studies described the results of ROC curve analyses (Appendix A). To predict subsequent MACE, the four included studies showed sensitivities ranging from 0.59 to 0.91. The specificities ranged from 0.69 to 0.85. For all four studies, the pooled parameters had a sensitivity of 0.80 (95% CI: 0.64–0.90) and a specificity of 0.78 (95% CI: 0.69–0.85) (Figure 2A,B). The summary ROC curve summarized the overall predictive accuracy, showing a tradeoff between sensitivity and specificity in which the calculated AUC was 0.84 (95% CI: 0.80–0.87) (Figure 2C).

### 3.4. Total Cardiac Events in Post-AMI Patients

In total, five studies provided adjusted RRs and 95% CIs for total cardiac events in post-AMI patients. There was moderate heterogeneity among these studies; therefore, a random-effects model was used (*I*^2^ = 64%; Chi-square= 11.17; *p* = 0.02). On combining all available studies, the RR analysis suggested a higher risk of total cardiac events in patients with abnormal MV than in patients with normal MV, with a mean follow-up of 36.4 months (*n* = 5; RR, 2.07; 95% CI: 1.28–3.37; *p* = 0.003) (Figure 3A).

### 3.5. Cardiac Death in Post-AMI Patients

Three studies provided adjusted RRs and 95% CIs for cardiac death in patients after AMI. There was moderate heterogeneity among these studies; therefore, a random-effects model was used (*I*^2^ = 71%; Chi-square = 6.85; *p* = 0.03). On combining all available studies, the RR analysis suggested a significantly higher risk of cardiac death in patients with abnormal MV than in patients with normal MV, with a mean follow-up of 43.7 months (*n* = 3; RR, 2.48; 95% CI: 1.03–5.96; *p* = 0.04) (Figure 3B).

### 3.6. Sensitivity Analysis

Sensitivity analysis was performed using the leave-one-out method for total cardiac events and cardiac death (Appendix A). The direction and magnitude of the pooled RRs did change with the removal of the study by Dwivedi et al. [16] (*I*^2^ changed from 64% to 0) (Appendix A) in the sensitivity analysis of total cardiac events, but they did not differ markedly with the removal of any of the other three studies (Appendix A). Similarly, when we conducted a sensitivity analysis for cardiac death, the direction and magnitude of the pooled RRs changed with the removal of the study by Khumri et al. [22] (*I*^2^ changed from 71% to 0) (Appendix A), but they did not differ markedly with the removal of any of the other two studies (Appendix A).

### 3.7. Meta-Analysis of the Studies That Reported HRs for Total Cardiac Events

A separate meta-analysis of four studies that reported HRs for total cardiac events was conducted (Figure 3C). There was moderate heterogeneity among these studies; therefore, a random-effects model was used (*I*^2^ = 66%; Chi-square = 8.73; *p* = 0.03). The HR analysis suggested a significantly higher risk of total cardiac events in patients with abnormal MV than in patients with normal MV (*n* = 4; HR: 2.09, 95% CI: 1.14–3.81; *p* = 0.02). This result is consistent with that of the meta-analysis of RRs (combining both HR and OR). We did not perform a separate meta-analysis on ORs for total cardiac events and cardiac death due to the limited number of studies.

### 3.8. Quality Assessment

All included studies had high scores (≥8) on the Newcastle–Ottawa scale assessment (Appendix A). Furthermore, all studies clearly defined the prognostic factor, provided appropriate cutoff points, and underwent measurements for adequate validity of the prognostic factor and adequacy of the analysis (Appendix A). The definition of the semiquantitative system in all studies was highly consistent, which was the core of this meta-analysis. Five of the six studies clearly indicated that all reviewers were blinded to the angiographic and clinical data. The sample size in two studies was small, although the results were still statistically significant on evaluating the adjusted HR. Publication bias was not evaluated using a funnel plot due to the small number of available studies (<10).

## 4. Discussion

This systematic review and meta-analysis demonstrated that MV detected by MCE is significantly associated with MACE after AMI, including total cardiac events (cardiac death, non-fatal myocardial infarction, and other cardiac events) and cardiac death. These findings support the argument that residual MV, as shown by low-power MCE, can be an independent predictor of MACE in post-AMI patients (*p* = 0.02) and can provide moderate predictive accuracy among these patients (AUC = 0.84). The ability to more accurately assess post-AMI patients is extremely important, as it may benefit therapeutic strategies, identify patients at potentially high risk for MACE, and provide closer observation, which may potentially improve clinical outcomes in post-AMI patients, even if most of them have already undergone percutaneous transluminal coronary angioplasty (PTCA) successfully [6].

With regard to risk stratification of AMI, it is important to evaluate resting left ventricular function, the degree and extent of residual MV, and myocardial ischemia [23]. MCE is a technique that can meet all these requirements [24]. MCE can detect contrast microbubbles at the capillary level within the myocardium. As 90% of the blood volume within the coronary circulation at rest in diastole is contained in the capillaries [25], MCE can evaluate the integrity of myocardial vasculature at the capillary level in real time [26]. Animal and clinical studies of AMI have shown that the size of the MCE defect correlate well with the size of the infarct. After AMI, collateral blood flow has been shown to be generally less than normal flow in areas showing myocardial viability. The MCE score index is calculated by adding the contrast scores of all segments and dividing them by the total number of segments. Thus, the MCE score index can represent the extent and degree of residual infarction, and thus it reflects the residual MV [24,27]. The recanalization of epicardial coronary arteries after reperfusion does not indicate the reopening of myocardial microvessels or the success of myocardial reperfusion (i.e., “no-reflow phenomenon”) [28]. This may be related to the destruction of myocardial microvascular structure and function, preventing the ischemic area from receiving adequate blood perfusion [11]. The identification of the degree and extent of the no-reflow phenomenon is of great value in predicting MV, cardiac function recovery, and the possibility of MACE after AMI [4,11,28].

A previous meta-analysis demonstrated that the sensitivity of MCE in detecting hibernating myocardium in patients with ischemic cardiomyopathy was similar to that of metabolic markers [29]. Studies [12,30] that compared MCE and SPECT in the same group demonstrated that MCE was superior to SPECT in predicting hard cardiac events or global recovery of function after AMI. Moreover, several studies [13,31,32,33] have shown that the performance of MCE was comparable to that of cardiac magnetic resonance imaging (CMR) in patients after AMI, and MCE had a comparable accuracy as a predictor of MACE in AMI patients. The 2017 European Association of Cardiovascular Imaging has already recommended MCE as a class IIA imaging method to improve MV evaluation [27]. This indicates that MCE is evolving as a valuable tool after AMI, and is a portable, ionizing radiation-free, bedside technique that allows for rapid examination and data acquisition by the clinician [24]. However, more evidence is needed to guide the clinical application of this technology. MCE may also be particularly useful in critical patients who cannot tolerate prolonged examinations such as CMR.

The heterogeneity of this meta-analysis decreased significantly when the studies by Dwivedi et al. [16] and Khumri et al. [22] were excluded from the sensitivity analysis for total cardiac events and cardiac death, respectively. Based on our analysis of these studies and data, we speculate that the reason for this heterogeneity may be that the treatment plans after AMI for the patients in these two studies were different from those in the other studies. In these two studies, there were fewer patients who underwent PTCA or only received medical therapy; in comparison, the majority of patients underwent PTCA in the other four studies. Furthermore, all patients in the studies by Abdelmoneim et al. [17] and Lenz et al. [13] achieved TIMI flow 3 after PCI. Previous studies [34,35,36,37] show that PCI leads to excellent clinical outcomes and a continued decline in perioperative adverse events [38]. Thus, patients who undergo PCI are more likely to be considered to have clinically lower-risk features than patients who undergo other treatments. According to the results of our sensitivity analysis, we may speculate that different treatments may affect the follow-up outcomes and cause heterogeneity in the meta-analysis and that the MCE index score may especially be useful in patients who have successfully undergone PTCA. This may also explain why the other three studies did not use cardiac death as an independent endpoint to estimate the prognostic power of MCE. This is because participants were at a relatively lower risk after PTCA, leading to a lower incidence of cardiac death and making statistical analysis a challenge. Therefore, we deduce that MV by MCE may be associated with a high risk for MACE in patients who have successfully undergone PTCA after AMI (for total cardiac events: RR = 2.58, 95% CI, 1.71–3.90; for cardiac event: RR = 4.10, 95% CI, 1.87–9.01).

According to our meta-analysis, SROC curve analysis revealed an AUC of 0.84, which indicated that the MCE score index could provide moderate predictive accuracy among patients after AMI. However, the MCE score index is only a semiquantitative parameter and not a continuous variable, limiting its clinical application in making an accurate diagnosis. MCE can be used for both qualitative and quantitative analysis. In previous studies, coronary flow reserve (CFR), which is one of the MCE quantitative parameters, was shown to correlate with MV and adverse ventricular remodeling after AMI [28,39,40]. A recent study [41] also demonstrated that CFR is an independent predictor of MACE in patients with AMI. Therefore, a large, prospective, multicenter study is necessary to definitively establish the MCE score index and other MCE quantitative parameters as long-term predictors of MACE after AMI.

This meta-analysis had some limitations. First, the limited number of studies and patients may limit the clinical potential of this technique. Second, the close association between MV and MACE seen in this meta-analysis indicates the need to develop and standardize more sophisticated parameters for quantifying MV after AMI to potentially improve the prognostic power and promote the clinical utility of this metric. Third, the MCE score index is a semi-quantitative parameter that depends on much of the training of the echo labs. Finally, the pooled relative RRs used to evaluate the results in this meta-analysis were calculated by combining HR and OR. Methodologically, it is common to perform this type of combined calculation, especially when the incidence of these endpoints is extremely low. To assess whether this model of analysis affected the results, we performed an additional analysis of the studies that reported HRs as the result. The results of the analysis of this subgroup’s HRs were consistent with those of the meta-analysis of pooled RRs.

## 5. Conclusions

Overall, residual MV assessment by MCE, which is an ionizing radiation-free and inexpensive technique, appears to have great long-term prognostic value in the prediction of MACE following AMI and can provide moderate predictive accuracy. The use of MCE at the bedside may help rapidly identify post-AMI patients who are at risk for MACE, allowing for the provision of optimal therapeutic strategies even if most of them have already undergone PTCA successfully.

## Figures and Tables

**Figure 1 medicina-58-01429-f001:**
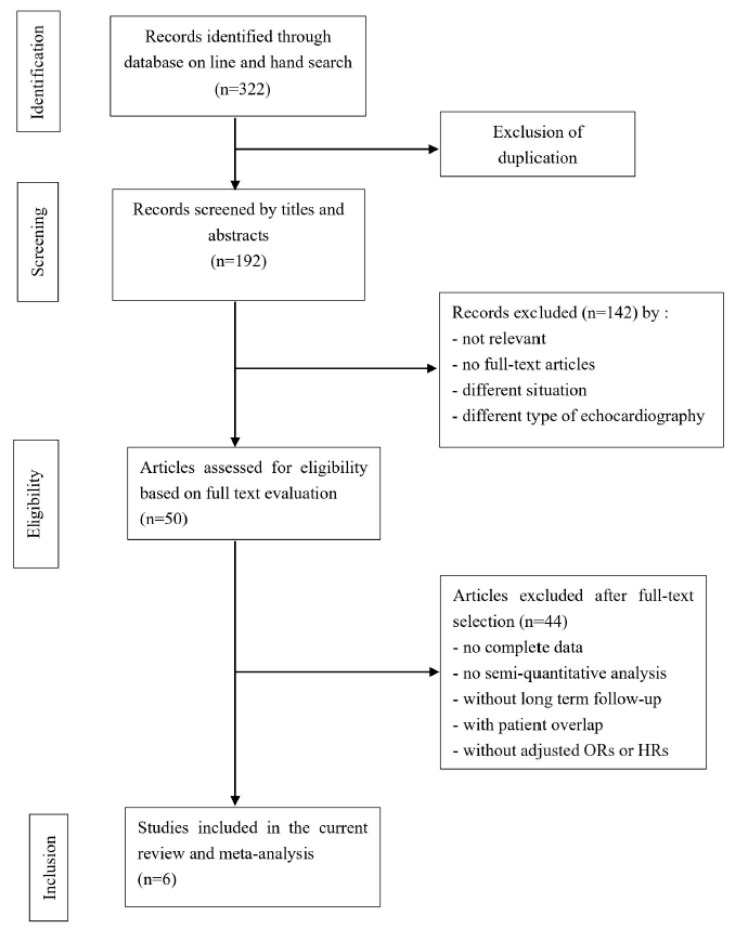
Flowchart of the study selection process.

**Figure 2 medicina-58-01429-f002:**
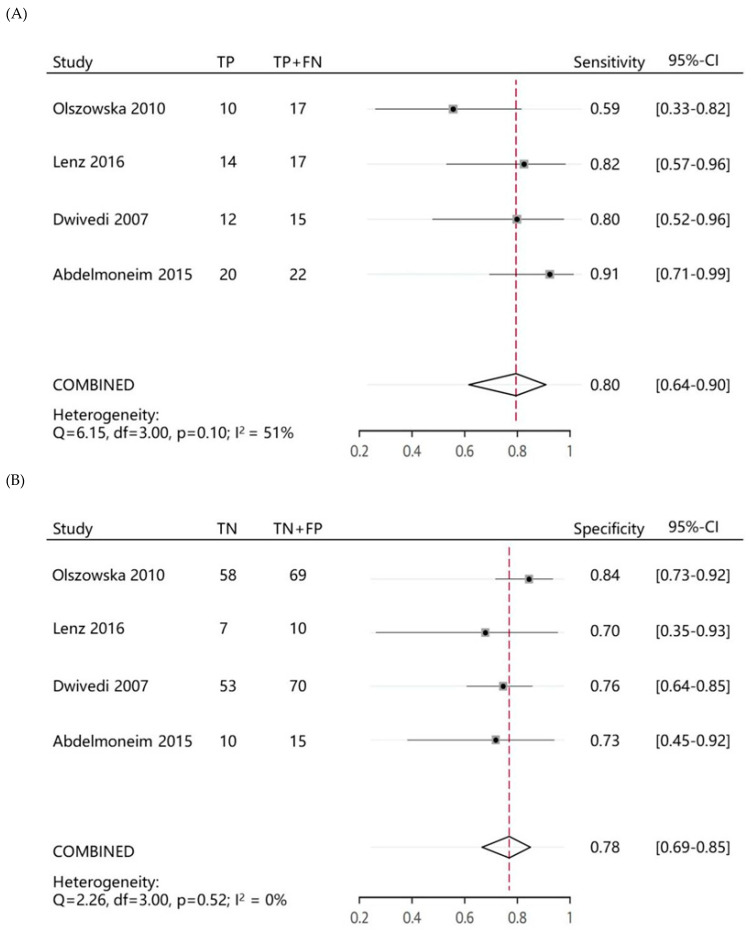
Forest plots of (**A**) pooled sensitivities, (**B**) specificities, and (**C**) summary receiver–operating characteristic curve (**C**) of myocardial contrast echocardiography to predict major adverse cardiac events during follow-up visits [13,14,16,17].

**Figure 3 medicina-58-01429-f003:**
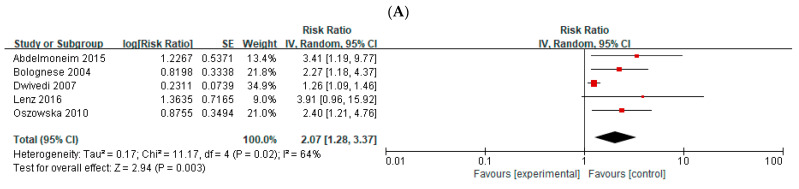
Forest plots of the pooled relative risks (RR) for (**A**) total cardiac events and (**B**) cardiac death, and (**C**) forest plot of pooled hazard ratios (HR) in patients after acute myocardial infarction [12,13,14,17,21,22].

**Table 1 medicina-58-01429-t001:** Summary of basic characteristics of selected studies for meta-analysis.

Study	Year	Patients (*n*)	Age (Years) Mean ± SD	Men (%)	DM (%)	HT (%)	Smoker	Hyperlipidemia	Family History of CAD	Follow-Up Years, ms
Bolognese et al. [21]	2004	124	62 ± 10	84	10	36	43	27	N/A	mean 46 ± 32
Khumri et al. [22]	2006	167	61.6 ± 14.7	61.1	18.0	56.9	57	34.1	N/A	mean 39
Dwivedi et al. [16]	2007	95	61 ± 10.8	75.8	26	32	42	44	27	mean 46 ± 16
Olszowska et al. [14]	2010	86	58.4 ± 11.2	79.1	26	60	44	77	36	mean 34 (range, 14–37)
Abdelmoneim et al. [17]	2015	37	64 ± 14	73.0	27	73	65	86	146	mean 14 ± 7 (median 17)
Lenz et al. [13]	2016	27	62 ± 13	70.4	29.6	74.1	70.4	85.2	55.6	mean 42 ± 31 (median 54)
Pooled		536								
**Study**	**Inclusion**		**Exclusion**					**Treatment**	**Primary Endpoints**	
Bolognese et al. [21]	(1) first AMI (2) successful primary PTCA within 6 h of the onset of symptoms or between 6 and 24 h if there was evidence of continuing ischemia		(1) IRA diameter stenosis 70% or inability to identify IRA, (2) HF or cardiogenic shock in the first week after infarction, (3) postinfarction angina, (4) significant other cardiac disease, (5) life-limiting noncardiac disease.					100% PTCA	MACE, including cardiac death, NFMI, hospitalization for congestive HF and combined events	
Khumri et al. [22]	(1) recent myocardial necrosis, (2) wall motion abnormalities, (3) coronary angiography before MCE.		(1) Patients with previous anterior wall AMI; (2) technically difficult to perform MCE					medical therapy	(1) all-cause mortality (2) a combined end point of mortality or heart failure	
Dwivedi et al. [16]	Patients shortly after their first presentation with AMI.		N/A					68% revascularization; 87% thrombolysis	(1) Cardiac death; (2) Cardiac death or NFMI	
Olszowska et al. [14]	Patients underwent PCI within 12 h of an anterior AMI.		Hemodynamically unstable patients					100% PCI	(1) Cardiac death (2) Other cardiac events: NFMI, recurrent angina with TLR or repeat hospitalization for congestive HF	
Abdelmoneim et al. [17]	Patients with evidence of STEMI.		N/A					84% coronary intervention 5% fibrinolytic therapy only	Any cardiac events including hard events (all-cause mortality; NFMI)	
Lenz et al. [13]	Patents with STEMI and TIMI III flow		N/A					85.2% coronary intervention	Any cardiac events: (1) hard events (all-cause mortality; NFMI) (2) soft events (development of typical angina; hospitalization for arrhythmias, chest pain, unstable angina or to rule out MI; cardiac revascularization; and/or development of HF)	

CAD, coronary artery disease; DM, diabetes mellitus; HT, hypertension; N/A, not available; AMI, Acute myocardial infarction; PTCA, percutaneous transluminal coronary angioplasty; IRA, infarct-related artery; MACE, major cardiac events N/A, not available; CD, Cardiac death; MI, myocardial infarction; STEMI, ST-segment elevation myocardial infarction; PCI, percutaneous coronary intervention; HF, heart failure; TLR, target lesion revascularization; PTCA, percutaneous transluminal coronary angioplasty; NFMI, non-fatal myocardial infarction.

**Table 2 medicina-58-01429-t002:** The number of cardiac events and Outcome of multivariate analysis of risk for cardiac event.

Study	The Number of Cardiac Events (*n*)		Multivariable RR	
	Total Cardiac Events	Cardiac Deaths	Total Cardiac Events	Cardiac Death
Bolognese L et al. [21]	14	9	OR, 2.27; 95% CI, 1.18–4.35, *p* = 0.01	OR, 3.85; 95% CI, 1.39–11.1, *p* = 0.01
Khumri TM et al. [22]	N/A	18	N/A	OR, 4.5; 95% CI, 1.3–15.4, *p* = 0.02
Dwivedi et al. [16]	15	8	HR, 1.26; 95% CI,1.09–1.44; *p* = 0.002	HR,1.37; 95% CI,1.08–1.75; *p* = 0.01
Olszowska et al. [14]	17	4	HR, 2.4; 95% CI, 1.21–2.6; *p* = 0.02	
Abdelmoneim et al. [17]	22	4	HR, 3.41; 95% CI,1.19–12.27; *p* = 0.020	
Lenz et al. [13]	23	6	HR, 3.91; 95% CI,0.96–21.8; *p* = 0.057	

RR = Risk ratio; HR: Hazard ratio; CI = Confidence interval.

**Table 3 medicina-58-01429-t003:** Summary of the characteristics of MCE and the Semiquantitative system.

Study	Characteristics of MCE			Semiquantitative System	
	No. of Segments	Contrast Agents	Time to Perform MCE	Definition of Semiquantitative Parameters	Semiquantitative Scoring System
Bolognese et al. [21]	16	iopamidol	shortly after PTCA	adding contrast scores of all segments and dividing by the total number of evaluable segments.	homogenous contrast perfusion = 2; partial/patchy contrast perfusion = 1; absent contrast perfusion = 0.
Khumri et al. [22]	16	Optison and Definity	a mean of 2 days (range 0 to 11) after AMI	a sum of the values for all interpretable segments divided by the number of segments analyzed	homogenous contrast perfusion = 1; partial/patchy contrast perfusion = 2; absent contrast perfusion = 3.
Dwivedi et al. [16]	16	40 patients, Optison; 55 patients, Sonovue	within 7 ± 2 days after AMI	Same as Dwivedi et al. [16]	Same as Khumri TM et al. [22]
Olszowska et al. [14]	16	Optison	within 5 ± 2 days after PCI	Same as Dwivedi et al. [16]	Same as Khumri TM et al. [22]
Abdelmoneim et al. [17]	17	Definity	within a mean (SD) of 29.3 (21) h of the CA	Same as Dwivedi et al. [16]	Same as Khumri TM et al. [22]
Lenz et al. [13]	17	Definity	within 1.04 ± 0.8 days after catheterization	Same as Dwivedi et al. [16]	homogenous contrast perfusion = 0; partial/patchy contrast perfusion = 1; absent contrast perfusion = 2.

MCE, myocardial contrast echocardiography; PTCA, percutaneous transluminal coronary angioplasty; PCI, percutaneous coronary intervention; CA, coronary angioplasty.

## Data Availability

Not applicable.

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
