# Peer review of "Long-Term Prognostic Value of Myocardial Viability by Myocardial Contrast Echocardiography in Patients after Acute Myocardial Infarction: A Systematic Review and Meta-Analysis"

_medicina, 2022, doi:10.3390/medicina58101429_

Round 1

Reviewer 1 Report

This is a systematic review/metaanalysis of studies documenting prognostic significance of myocardial viability detected by MCE with regard to MACE, including a total of 536 patients.  The concept is interesting, though there are some points that should be addressed.

Major points

1. The concept of assessment of myocardial viability using MCE should be discussed in detail, since it is the crucial aspect of this study. The authors should explain what it is that the MCE really measures, what is the physiological significance of contrast defect/score index. What is the relation between myocardial perfusion and viability.

It is not enough to just state: “However, myocardial contrast echocardiography (MCE) can reflect the condition of myocardial microcirculation by myocardial per-70 fusion, which can evaluate the residual myocardial viability (MV).”

2. The figures for the manuscript are missing.

Minor points:

1. First paragraph of “Results” seems to be an instruction that should be deleted.

Author Response

Dear Professor:

We do appreciate you for the positive view of our work. Those comments are valuable and very helpful for improving the quality of our manuscript. According to the recommendations from you, we have carefully amended the manuscript and answered the questions point-by-point. All the requirements or comments were answered or stressed as follows:

  1. The concept of assessment of myocardial viability using MCE should be discussed in detail, since it is the crucial aspect of this study. The authors should explain what it is that the MCE really measures, what is the physiological significance of contrast defect/score index. What is the relation between myocardial perfusion and viability.

It is not enough to just state: “However, myocardial contrast echocardiography (MCE) can reflect the condition of myocardial microcirculation by myocardial per-70 fusion, which can evaluate the residual myocardial viability (MV).”

Response: Thanks for your suggestion, which is very meaningful! MCE can detect contrast microbubbles at the capillary level within the myocardium [1]. When the concentration of myocardial microbubbles reaches a steady state during a contrast administration, the signal intensity can provide a measure of the myocardial blood volume fraction, which was mainly comprised by capillary blood. Thus, MCE image can provide the assessment of capillary density in the different myocardial regions[1]. Myocardial blood perfusion refers to the blood flow in the capillaries. Contrast signal intensity during MCE has been shown to correlate directly with capillary density in patients with chronic heart failure[2] . Animal studies of AMI have shown that the size of the MCE defect assessed about 15 seconds after administration, correlate well with the size of the infarct. [3]. This conclusion was confirmed in patients following AMI[4] . After AMI, collateral blood flow has been shown to be generally less than normal flow in areas showing myocardial viability [5]. Regions with normal blood flow fill within 5 s after destruction imaging. Regions with collateral flow fill later depending on the magnitude of flow. If they do not fill within 15 s, then flow to the region is markedly reduced and will result in necrosis [6]. This method of assessment of myocardial viability by MCE has been shown to predict recovery of regional function and transmural extent of infarction, and also has been shown to predict recovery of both regional and global LV function [7-9].

The semiquantitative scoring system that used to assess contrast intensity after microbubble destruction, was divided into a three-point scale: homogenous contrast perfusion, partial/patchy contrast perfusion, and absent contrast perfusion (line 184-186). MCE score index, as mentioned in this meta- analysis (line 181-183) was obtained by adding MCE scores of all segments and dividing by the total number of evaluable segments. The MCE score index represented the extent and intensity of residual infarction and thus residual myocardial viability. Percentages of segments with heterogeneous perfusion (patchy infarction) and absent contrast perfusion (complete infarction) also were obtained for each patient.

The related revision had been done in the manuscript.

References

[1] Kaul S, Jayaweera AR. Coronary and myocardial blood volumes: noninvasive tools to assess the coronary microcirculation?[J]. Circulation, 1997, 96(3): 719-724.

[2] Shimoni S, Frangogiannis NG, Aggeli CJ, et al. Microvascular structural correlates of myocardial contrast echocardiography in patients with coronary artery disease and left ventricular dysfunction: implications for the assessment of myocardial hibernation[J]. Circulation, 2002, 106(8): 950-956.

[3] Sieswerda GT, Yang L, Boo MB, et al. Real-time perfusion imaging: a new echocardiographic technique for simultaneous evaluation of myocardial perfusion and contraction[J]. Echocardiography, 2003, 20(6): 545-555.

[4] Swinburn JM, Lahiri A, Senior R. Intravenous myocardial contrast echocardiography predicts recovery of dysynergic myocardium early after acute myocardial infarction[J]. J Am Coll Cardiol, 2001, 38(1): 19-25.

[5] Janardhanan R, Burden L, Senior R. Usefulness of myocardial contrast echocardiography in predicting collateral blood flow in the presence of a persistently occluded acute myocardial infarction-related coronary artery[J]. Am J Cardiol, 2004, 93(10): 1207-1211.

[6] Coggins MP, Sklenar J, Le DE, et al. Noninvasive prediction of ultimate infarct size at the time of acute coronary occlusion based on the extent and magnitude of collateral-derived myocardial blood flow[J]. Circulation, 2001, 104(20): 2471-2477.

[7] Janardhanan R, Swinburn JM, Greaves K, et al. Usefulness of myocardial contrast echocardiography using low-power continuous imaging early after acute myocardial infarction to predict late functional left ventricular recovery[J]. Am J Cardiol, 2003, 92(5): 493-497.

[8] Janardhanan R, Moon JC, Pennell DJ, et al. Myocardial contrast echocardiography accurately reflects transmurality of myocardial necrosis and predicts contractile reserve after acute myocardial infarction[J]. Am Heart J, 2005, 149(2): 355-362.

[9] Main ML, Magalski A, Kusnetzky LL, et al. Usefulness of myocardial contrast echocardiography in predicting global left ventricular functional recovery after anterior wall acute myocardial infarction[J]. Am J Cardiol, 2004, 94(3): 340-342.

  1. The figures for the manuscript are missing.

 Response: Thanks for your advice! We carefully rechecked the figures in the article again.

Minor points:

  1. First paragraph of “Results” seems to be an instruction that should be deleted.

Response: Thanks for your suggestion! We just wonder whether the “first paragraph” that you mentioned refers to the “Abstract” of the paper? Or the “Introduction” part?

Now we would like to express our most sincere gratitude for your extraordinary help to polish our work. Thanks again for giving us this chance to make the substantial revisions according to your comments. We hope the above responses can address your questions properly and please don’t hesitate to contact us if you have any further questions. We will try our best to reply.

Best regards,

Ying-kun Guo, MD

Reviewer 2 Report

Dear authors

Regarding the manuscript, I point the following:

line 35: "according to recent guidelines, myocardial contrast echocardiography 34 (MCE) is recommended for detecting residual myocardial viability (MV)." References in the text are necessary.

line 37: "However, the long-term 35 prognostic value of MV as assessed by MCE in identifying major adverse cardiac events (MACE) 36 after acute myocardial infarction (AMI) remains undefined. " References in the text are necessary.

line 52: "... in clinical practice, 51 especially for patients who cannot tolerate prolonged examinations. " This is unclear and the sentence must be better written to explain it.

line 63: "Therefore, risk stratification in patients after AMI is crucial to guide treatment strategies 62 and improve long-term outcomes.5, 6" There are clinical essays pointing out this with risk stratification made for different imaging techniques including stress echo and contrast echo.

line 68: "... However, even among patients with thrombolysis in myocardial infarction 66 (TIMI) flow grade 3 after treatment, there are still some with myocardial perfusion defects, 67 such as a no-reflow or low-reflow phenomenon.". This is a short running sentence, not well written and is totally unclear. 

line 98: "in- 96 cluding “myocardial,” “contrast echocardiography,” “infarct*,” “event*,” ‘‘prognosis*,” 97 Medicina 2022, 58, x FOR PEER REVIEW 3 of 10 “predict*,” and ‘‘diagnosis*.’’ It is missing here the term "viability" and this can compromise the search.

line 101: "The search analyzed original literature published in 99 such databases up to June 1, 2022, with no date restrictions. " Why only up to 20 years ago?

line 105: "... post-AMI patients. ..."What type of AMI? Did the authors include non-ST segment elevation AMI patients or only ST-segment elevations AMI? 

line 107: "The inclusion criteria in this meta-analysis were as follows: (1) prospective study, (2) 104 population: post-AMI patients, (3) use of MCE to assess MV, (4) semiquantitative scoring 105 of MCE findings, (5) at least 6 months follow-up, and (6) full-text in the English language. 106 The endpoint was MACE, which included total cardiac events and cardiac death. "The reviewer points out that the gold standard criteria for the diagnosis of myocardial viability is not the myocardial perfusion with agent contrats and echocardiography.

 line 162: "At last, six studies were eventually used for a detailed 162 study. 13, 14, 16, 17, 21, 22 A flowchart of our search results is presented in Fig. 1." This is such an anecdotal very small number of studies under analysis.

Line 167 : "In total, there were 536 patients across the studies, ranging from 27 to 167 pa- 165 tients per study. Demographically, 73.8% of the subjects were male, and their ages ranged 166 from 47 to 78 years. "

Line 178: "ach study named the semiquantitative prognostic parameter differently, including 176 contrast defect index (CDI), contrast score index (CSI), regional perfusion score index 177 (RPSI), and perfusion score index (PSI). " The observed lack of standardization in the diagnosis of viability by myocardial contrast echocardiography makes extremely troublesome the analysis and the reviewer points out that this is done always on a semiquantative basis that depends on much of the training of the echo labs.

line  188: "To predict subsequent MACE, the 187 four included studies showed sensitivities ranging from 0.59 to 0.91. The specificities 188 ranged from 0.69 to 0.85." How do the authors explain such a wide range of sensitivities and specificities?

line 197: "There was moderate heterogeneity among these studies; ..." Only moderate?

line 242: "Five of the six studies clearly indicated that all 238 reviewers were blinded to the angiographic and clinical data. The sample size in two stud- 239 ies was small, although the results were still statistically significant on evaluating the ad- 240 justed HR. Publication bias was not evaluated using a funnel plot due to the small number 241 of available studies (< 10)." There is significant bias based on the present manuscript as pointed out by the authors.

line 254: "The ability to more accurately 249 assess post-AMI patients is extremely important, as it may benefit therapeutic strategies, 250 identify patients at potentially high risk for MACE, and provide closer observation, which 251 may potentially improve clinical outcomes in post-AMI patients, even if most of them 252 have already undergone percutaneous transluminal coronary angioplasty (PTCA) suc- 253 cessfully. 6 254 Medicina 2022, 58, x FOR PEER REVIEW". The reviewer agrees with this sentence but sustains that this is under serious scrutiny in the literature considering that there are already gold standards for myocardial viability identification.

line 237: "The definition of the semiquantitative system in all studies was highly consistent, 237 which was the core of this meta-analysis." Where is the quality control of the reviewers of the myocardial contrast echo laboratories that were the basis of publications of such different and widespread studies?

line 256: "MCE 256 is a technique that can meet all these requirements. 24". The mentioned reference is an opinion article and not an original state-of-the-art randomized clinical trial.

line 280: "edside technique that allows for rapid examination and data acquisition by the 279 clinician.24 However, more evidence is needed to guide the clinical application of this tech- 280 nology. " This is probably the most important conclusion of the authors.

line 285: "The heterogeneity of this meta-analysis decreased significantly when the studies by 283 Dwivedi et al.16 and Khumri et al.22 were excluded from the sensitivity analysis for total 284 cardiac events and cardiac death, respectively." The reviewer asks if the authors drew a conclusion subtracting these two studies. The reviewer supposes that not, because if the present carries on substantial doubts including because it is based on such a small number of publications it should possibly be more difficult if these two publications come to be excluded.

line 297: "According to the results of our sensitivity 294 analysis, we may speculate that different treatments may affect the follow-up outcomes 295 and cause heterogeneity in the meta-analysis and that the MCE index score may especially 296 be useful in patients who have successfully undergone PTCA." The reviewer would to see this study.

line 325: "This meta-analysis had some limitations. First, the limited number of studies and pa- 315 tients may limit the clinical potential of this technique. Second, the close association be- 316 tween MV and MACE seen in this meta-analysis indicates the need to develop and stand- 317 ardize more sophisticated parameters for quantifying MV after AMI to potentially im- 318 prove the prognostic power and promote the clinical utility of this metric. Finally, the 319 pooled relative RRs used to evaluate the results in this meta-analysis were calculated by 320 combining HR and OR. Methodologically, it is common to perform this type of combined 321 calculation, especially when the incidence of these endpoints is extremely low. To assess 322 whether this model of analysis affected the results, we performed an additional analysis 323 of the studies that reported HRs as the result. The results of the analysis of this subgroup’s 324 HRs were consistent with those of the meta-analysis of pooled RRs."  The reviewer points out that there are unsolved and unanswered limitations of the present manuscript.

Author Response

Dear Professor:

We do appreciate you for the positive view of our work. Those comments are valuable and very helpful for improving the quality of our manuscript. According to the recommendations from you, we have carefully amended the manuscript and answered the questions point-by-point. All the requirements or comments were answered or stressed as follows:

  1. line 35: "according to recent guidelines, myocardial contrast echocardiography 34 (MCE) is recommended for detecting residual myocardial viability (MV)." References in the text are necessary.

Response: Thanks for your suggestion! Since this is the abstract of the paper, we did not insert references. The corresponding reference here is below, which is also the NO.25 in the Index of references that at the end of our article.

“25. Senior R, Becher H, Monaghan M, et al. Clinical practice of contrast echocardiography: recommendation by the European Association of Cardiovascular Imaging (EACVI) 2017. Eur Heart J Cardiovasc Imaging 2017;18:1205-05af.”

  1. line 37: "However, the long-term 35 prognostic value of MV as assessed by MCE in identifying major adverse cardiac events (MACE) 36 after acute myocardial infarction (AMI) remains undefined. " References in the text are necessary.

Response: Since this is the abstract of the paper, we did not insert references. The corresponding reference here is below, which is also the NO.13/14/16/17/21/22/ in the Index of references that at the end of our article.

“13. Lenz CJ, Abdelmoneim SS, Anavekar NS, et al. A comparison of infarct mass by cardiac magnetic resonance and real time myocardial perfusion echocardiography as predictors of major adverse cardiac events following reperfusion for ST elevation myocardial infarction. Echocardiography 2016;33:1539-45.

  1. Olszowska M, Kostkiewicz M, Podolec P, et al. Myocardial viability detected by myocardial contrast echocar-diography--prognostic value in patients after myocardial infarction. Echocardiography 2010;27:430-4.
  2. Dwivedi G, Janardhanan R, Hayat SA, et al. Prognostic value of myocardial viability detected by myocardial contrast echocardiography early after acute myocardial infarction. J Am Coll Cardiol 2007;50:327-34.
  3. Abdelmoneim SS, Martinez MW, Mankad SV, et al. Resting qualitative and quantitative myocardial contrast echocardiography to predict cardiac events in patients with acute myocardial infarction and percutaneous revascu-larization. Heart and vessels 2015;30:45-55.
  4. Bolognese L, Carrabba N, Parodi G, et al. Impact of microvascular dysfunction on left ventricular remodeling and long-term clinical outcome after primary coronary angioplasty for acute myocardial infarction. Circulation 2004;109:1121-6.
  5. Khumri TM, Nayyar S, Idupulapati M, et al. Usefulness of myocardial contrast echocardiography in predicting late mortality in patients with anterior wall acute myocardial infarction. Am J Cardiol 2006;98:1150-5.”

  1. line 52: "... in clinical practice, 51 especially for patients who cannot tolerate prolonged examinations. " This is unclear and the sentence must be better written to explain it.

Response: Thanks for your suggestion! We are sorry for the unclear of description. We rewrote the sentence as below: “…especially for patients who cannot tolerate those examinations that take a long time to complete.” The related revision had been done in the manuscript.

  1. line 63: "Therefore, risk stratification in patients after AMI is crucial to guide treatment strategies 62 and improve long-term outcomes.5, 6 " There are clinical essays pointing out this with risk stratification made for different imaging techniques including stress echo and contrast echo.

Response: Thanks for your suggestion! As you mentioned, there are many imaging examinations that can be used to do risk stratification after AMI. And they each have their own advantages and disadvantages. For example, DSE may be less sensitive than techniques that assess microvasculature (MCE) for the detection of hibernating myocardium as MBF reserve may be significantly reduced but the microvasculature may be intact[1]. Unlike DSE, MCE can detect myocardial viability at rest, and unlike radionuclide perfusion imaging, MCE has no radiation burden and can be performed at the bedside. MCE also has many limitations. Contrast intensity may be reduced at the bases of the heart, because the ultrasound power is weakest in the far field, thereby giving rise to false perfusion defects. Although recent advancements in technology and understanding of the interaction between microbubbles and ultrasound have improved both the imaging and technical issues.

[1] Hickman M, Chelliah R, Burden L, et al. Resting myocardial blood flow, coronary flow reserve, and contractile reserve in hibernating myocardium: implications for using resting myocardial contrast echocardiography vs. dobutamine echocardiography for the detection of hibernating myocardium[J]. Eur J Echocardiogr, 2010, 11(9): 756-762.

  1. line 68: "... However, even among patients with thrombolysis in myocardial infarction 66 (TIMI) flow grade 3 after treatment, there are still some with myocardial perfusion defects, 67 such as a no-reflow or low-reflow phenomenon." This is a short running sentence, not well written and is totally unclear. 

Response: We are sorry for the unclear of description. We rewrote the sentence as below: “However, even with a successful recanalization of criminal vessels, part of patients still suffer unsuccessful myocardial reperfusion. This phenomenon is known as no-reflow or low-reflow.” The related revision had been done in the manuscript.

  1. line 98: "in- 96 cluding “myocardial,” “contrast echocardiography,” “infarct*,” “event*,” ‘‘prognosis*,” 97 Medicina 2022, 58, x FOR PEER REVIEW 3 of 10 “predict*,” and ‘‘diagnosis*.’’ It is missing here the term "viability" and this can compromise the search.

Response: Thanks for your suggestion! We didn't use the term "viability" in the Search Strategy. This is in order to expand the search scope as possible and avoid the possibility of missing relevant literatures.

  1. line 101: "The search analyzed original literature published in 99 such databases up to June 1, 2022, with no date restrictions. " Why only up to 20 years ago?

Response: Actually, there is no date restrictions for our literature search,as mentioned in line 99-100.

  1. line 105: "... post-AMI patients. ..."What type of AMI? Did the authors include non-ST segment elevation AMI patients or only ST-segment elevations AMI? 

Response: Thanks for your suggestion! We are sorry for the unclear of expression. All patients included were ST-segment elevations AMI. The related revision had been done in the manuscript.

  1. line 107: "The inclusion criteria in this meta-analysis were as follows: (1) prospective study, (2) 104 population: post-AMI patients, (3) use of MCE to assess MV, (4) semiquantitative scoring 105 of MCE findings, (5) at least 6 months follow-up, and (6) full-text in the English language. 106 The endpoint was MACE, which included total cardiac events and cardiac death. "The reviewer points out that the gold standard criteria for the diagnosis of myocardial viability is not the myocardial perfusion with agent contrats and echocardiography.

Response: Thanks for your suggestion! We are sorry for the unclear of expression. What we want to express was that this meta-analysis only focuses on studies using MCE. The related revision had been done in the manuscript.

  1. line 162: "At last, six studies were eventually used for a detailed 162 study. 13, 14, 16, 17, 21, 22 A flowchart of our search results is presented in Fig. 1." This is such an anecdotal very small number of studies under analysis.

Response: Sorry for that. But We already had done as thorough a literature search as possible.

  1. Line 167 : "In total, there were 536 patients across the studies, ranging from 27 to 167 pa- 165 tients per study. Demographically, 73.8% of the subjects were male, and their ages ranged 166 from 47 to 78 years. "

Response: Sorry for that. But We already had done as thorough a literature search as possible.

  1. Line 178: "ach study named the semiquantitative prognostic parameter differently, including 176 contrast defect index (CDI), contrast score index (CSI), regional perfusion score index 177 (RPSI), and perfusion score index (PSI). " The observed lack of standardization in the diagnosis of viability by myocardial contrast echocardiography makes extremely troublesome the analysis and the reviewer points out that this is done always on a semiquantative basis that depends on much of the training of the echo labs.

Response: Thanks for your suggestion! We added this point in limitation. However, the rule of this semi-quantitative parameter, that is a three-point scale ( just homogenous, absent and partial contrast perfusion) is not so difficult for most sonographers with specialized training. Just as the first perfusion sequence and late gadolinium enhancement of CMR are also semi-quantitative analyses, which have been already widely used in clinical practice and play an important role in clinical diagnosis. Besides, the experienced reader blinded to the clinical and angiographic details performed analysis of echocardiographic data. The MCE and LV function data were assessed separately. These also contribute to the objectivity of the data.

  1. line 188: "To predict subsequent MACE, the 187 four included studies showed sensitivities ranging from 0.59 to 0.91. The specificities 188 ranged from 0.69 to 0.85." How do the authors explain such a wide range of sensitivities and specificities?

Response: The data mentioned above are the original data from the literature included in the meta-analysis. It is just because the results of these literature were inconsistent that meta-analysis is necessary. For all studies in our meta-analysis, the pooled parameters had a sensitivity of 0.80 (95% CI: 0.64–0.90) and a specificity of 0.78 (95% CI: 0.69–0.85). (line 192-196)

  1. line 197: "There was moderate heterogeneity among these studies; ..." Only moderate?

Response: In the methodology of meta-analysis, one of the common methods to evaluating heterogeneity is I2. When I2 is in the range of 50%-75%, moderate heterogeneity is indicated. In our meta-analysis, In our study, the I2 of all results was less than 75%, indicating heterogeneity as moderate.

15, line 242: "Five of the six studies clearly indicated that all 238 reviewers were blinded to the angiographic and clinical data. The sample size in two stud- 239 ies was small, although the results were still statistically significant on evaluating the ad- 240 justed HR. Publication bias was not evaluated using a funnel plot due to the small number 241 of available studies (< 10)." There is significant bias based on the present manuscript as pointed out by the authors.

Response: In the meta-analysis, the analysis of publication bias is not required if less than 10 original literatures are included.

  1. line 254: "The ability to more accurately 249 assess post-AMI patients is extremely important, as it may benefit therapeutic strategies, 250 identify patients at potentially high risk for MACE, and provide closer observation, which 251 may potentially improve clinical outcomes in post-AMI patients, even if most of them 252 have already undergone percutaneous transluminal coronary angioplasty (PTCA) suc- 253 cessfully. 6 254 Medicina 2022, 58, x FOR PEER REVIEW". The reviewer agrees with this sentence but sustains that this is under serious scrutiny in the literature considering that there are already gold standards for myocardial viability identification.

Response: Thanks for your suggestion! As you mentioned, there are many imaging examinations that can be used to do myocardial viability identification of AMI. And they each have their own advantages and disadvantages. For example, DSE may be less sensitive than techniques that assess microvasculature (MCE) for the detection of hibernating myocardium as MBF reserve may be significantly reduced but the microvasculature may be intact[1]. Unlike DSE, MCE can detect myocardial viability at rest, and unlike radionuclide perfusion imaging, MCE has no radiation burden and can be performed at the bedside. MCE also has many limitations. Contrast intensity may be reduced at the bases of the heart, because the ultrasound power is weakest in the far field, thereby giving rise to false perfusion defects. However, recent advancements in technology and understanding of the interaction between microbubbles and ultrasound have improved both the imaging and technical issues.

[1] Hickman M, Chelliah R, Burden L, et al. Resting myocardial blood flow, coronary flow reserve, and contractile reserve in hibernating myocardium: implications for using resting myocardial contrast echocardiography vs. dobutamine echocardiography for the detection of hibernating myocardium[J]. Eur J Echocardiogr, 2010, 11(9): 756-762.

  1. line 237: "The definition of the semiquantitative system in all studies was highly consistent, 237 which was the core of this meta-analysis." Where is the quality control of the reviewers of the myocardial contrast echo laboratories that were the basis of publications of such different and widespread studies?

Response: As mentioned above, meta-analysis cannot directly control the quality of the original data of literatures. However, the rule of this semi-quantitative parameter, that is a three-point scale ( just homogenous, absent and partial contrast perfusion) is not so difficult for most sonographers with specialized training. Besides, the experienced reader blinded to the clinical and angiographic details performed analysis of echocardiographic data. The MCE and LV function data were assessed separately. These also contribute to the objectivity of the data.

  1. line 256: "MCE 256 is a technique that can meet all these requirements. 24". The mentioned reference is an opinion article and not an original state-of-the-art randomized clinical trial.

Response: Thanks for your suggestion! We are sorry for the unclear of expression. In fact, this is not based on a single clinical trial, but the in understanding of MCE is based on the results of several animal and clinical experiments in recent years. Animal studies have shown that MCE defect size assessed 10-15 s after contrast administration, corresponded to infarct size[1,2]. This was confirmed in patients following AMI [3]. The extent and intensity of contrast defect and the magnitude of resting MBF reduction predicted the transmural extent of myocardial necrosis assessed by late gadolinium CMR imaging[4,5]. The ability of MCE to predict functional recovery is comparable to that of cardiac MRI[5]. Studies have also shown that among all the clinical, ECG and angiographic parameters of reperfusion after AMI, contrast perfusion is the only independent predictor of reperfusion[6-8].

[1] Coggins MP, Sklenar J, Le DE, et al. Noninvasive prediction of ultimate infarct size at the time of acute coronary occlusion based on the extent and magnitude of collateral-derived myocardial blood flow[J]. Circulation, 2001, 104(20): 2471-2477.

[2] Lafitte S, Higashiyama A, Masugata H, et al. Contrast echocardiography can assess risk area and infarct size during coronary occlusion and reperfusion: experimental validation[J]. J Am Coll Cardiol, 2002, 39(9): 1546-1554.

[3] Swinburn JM, Lahiri A, Senior R. Intravenous myocardial contrast echocardiography predicts recovery of dysynergic myocardium early after acute myocardial infarction[J]. J Am Coll Cardiol, 2001, 38(1): 19-25.

[4] Janardhanan R, Moon JC, Pennell DJ, et al. Myocardial contrast echocardiography accurately reflects transmurality of myocardial necrosis and predicts contractile reserve after acute myocardial infarction[J]. Am Heart J, 2005, 149(2): 355-362.

[5] Choi EY, Seo HS, Park S, et al. Prediction of transmural extent of infarction with contrast echocardiographically derived index of myocardial blood flow and myocardial blood volume fraction: comparison with contrast-enhanced magnetic resonance imaging[J]. J Am Soc Echocardiogr, 2006, 19(10): 1211-1219.

[6] Greaves K, Dixon SR, Fejka M, et al. Myocardial contrast echocardiography is superior to other known modalities for assessing myocardial reperfusion after acute myocardial infarction[J]. Heart, 2003, 89(2): 139-144.

[7] Bolognese L, Carrabba N, Parodi G, et al. Impact of microvascular dysfunction on left ventricular remodeling and long-term clinical outcome after primary coronary angioplasty for acute myocardial infarction[J]. Circulation, 2004, 109(9): 1121-1126.

[8] Galiuto L, Garramone B, Scarà A, et al. The extent of microvascular damage during myocardial contrast echocardiography is superior to other known indexes of post-infarct reperfusion in predicting left ventricular remodeling: results of the multicenter AMICI study[J]. J Am Coll Cardiol, 2008, 51(5): 552-559.

  1. line 280: "edside technique that allows for rapid examination and data acquisition by the 279 clinician.24 However, more evidence is needed to guide the clinical application of this tech- 280 nology. " This is probably the most important conclusion of the authors.

Response: Thanks for your comments. Actually, a previous meta-analysis demonstrated that the sensitivity of MCE in detecting hibernating myocardium in patients with ischemic cardiomyopathy was similar to that of metabolic markers[1]. With accumulating evidence of its prognostic value for the detection of myocardial viability over and above clinical markers and LVEF[2,3], MCE is evolving as a useful bedside technique for the assessment of myocardial viability.

[1] Shah BN, Khattar RS, Senior R. The hibernating myocardium: current concepts, diagnostic dilemmas, and clinical challenges in the post-STICH era[J]. Eur Heart J, 2013, 34(18): 1323-1336.

[2] Khumri TM, Nayyar S, Idupulapati M, et al. Usefulness of myocardial contrast echocardiography in predicting late mortality in patients with anterior wall acute myocardial infarction[J]. Am J Cardiol, 2006, 98(9): 1150-1155.

[3] Dwivedi G, Janardhanan R, Hayat SA, et al. Prognostic value of myocardial viability detected by myocardial contrast echocardiography early after acute myocardial infarction[J]. J Am Coll Cardiol, 2007, 50(4): 327-334.

  1. line 285: "The heterogeneity of this meta-analysis decreased significantly when the studies by 283 Dwivedi et al.16 and Khumri et al.22 were excluded from the sensitivity analysis for total 284 cardiac events and cardiac death, respectively." The reviewer asks if the authors drew a conclusion subtracting these two studies. The reviewer supposes that not, because if the present carries on substantial doubts including because it is based on such a small number of publications it should possibly be more difficult if these two publications come to be excluded.

Response: This meta-analysis did not exclude the above literatures in the data analysis to draw a conclusion. We just used sensitivity analysis to try to analyze the sources of heterogeneity. And sensitivity analysis was performed using the leave-one-out method, which was a common method for such analysis. The corresponding discussion were in line 293-314.

  1. line 297: "According to the results of our sensitivity 294 analysis, we may speculate that different treatments may affect the follow-up outcomes 295 and cause heterogeneity in the meta-analysis and that the MCE index score may especially 296 be useful in patients who have successfully undergone PTCA." The reviewer would to see this study.

Response: This is not a conclusion based on any reference, but the possible cause of heterogeneity inferred from the results of sensitivity analysis of the original literature data. And also it is not the main conclusion of this meta-analysis.

  1. line 325: "This meta-analysis had some limitations. First, the limited number of studies and pa- 315 tients may limit the clinical potential of this technique. Second, the close association be- 316 tween MV and MACE seen in this meta-analysis indicates the need to develop and stand- 317 ardize more sophisticated parameters for quantifying MV after AMI to potentially im- 318 prove the prognostic power and promote the clinical utility of this metric. Finally, the 319 pooled relative RRs used to evaluate the results in this meta-analysis were calculated by 320 combining HR and OR. Methodologically, it is common to perform this type of combined 321 calculation, especially when the incidence of these endpoints is extremely low. To assess 322 whether this model of analysis affected the results, we performed an additional analysis 323 of the studies that reported HRs as the result. The results of the analysis of this subgroup’s 324 HRs were consistent with those of the meta-analysis of pooled RRs."  The reviewer points out that there are unsolved and unanswered limitations of the present manuscript.

Response: Thank you so much for your suggestion! We added some points to our limitation section. The related revision had been done in the manuscript.

Now we would like to express our most sincere gratitude for your extraordinary help to polish our work. Thanks again for giving us this chance to make the substantial revisions according to your comments. We hope the above responses can address your questions properly and please don’t hesitate to contact us if you have any further questions. We will try our best to reply.

Best regards,

Ying-kun Guo, MD

Reviewer 3 Report

Wang and colleagues present an interesting systematic assessment of contrasto echo Cardio grappò in the prognostic assessment of patients with myocardial infarction.

The work is well written, the analysis carefully performed, and the data clearly presented.

I only have some minor comments: could You please clarify at which time point after myocardial infarction was contrast echocardiography performed?

Did any author report on the relationship between myocardial viabilità and risk of ventricular arrhythmias?

Author Response

Dear Professor:

We do appreciate you for the positive view of our work. Those comments are valuable and very helpful for improving the quality of our manuscript. According to the recommendations from you, we have carefully amended the manuscript and answered the questions point-by-point. All the requirements or comments were answered or stressed as follows:

  1. could You please clarify at which time point after myocardial infarction was contrast echocardiography performed?

Response: We are sorry for the unclear of description. In all studies, MCE was completed within a short time after AMI, ranging from 1.04±0.8 days to 7±2 days.

  1. Did any author report on the relationship between myocardial viabilità and risk of ventricular arrhythmias?

Response: In all the original literatures included in our meta-analysis, no author mentioned the relationship between myocardial viability and risk of ventricular arrhythmias.

Now we would like to express our most sincere gratitude for your extraordinary help to polish our work. We hope the above responses can address your questions properly and please don’t hesitate to contact us if you have any further questions. We will try our best to reply.

Best regards,

Ying-kun Guo, MD